# Activity-Dependent Non-Coding RNA MAPK Interactome of the Human Epileptic Brain

**DOI:** 10.3390/ncrna9010003

**Published:** 2023-01-04

**Authors:** Allison Kirchner, Fabien Dachet, Leonard Lipovich, Jeffrey A. Loeb

**Affiliations:** 1Department of Neurology and Rehabilitation, University of Illinois at Chicago, 912 S Wood Street, Chicago, IL 60612, USA; 2University of Illinois NeuroRepository, University of Illinois at Chicago, 912 S Wood Street, Chicago, IL 60612, USA; 3Department of Basic Medical Sciences, College of Medicine, Mohammed Bin Rashid University of Medicine and Health Sciences, Dubai 505055, United Arab Emirates

**Keywords:** MAPK signaling, epilepsy, interactome, evolution, lncRNA

## Abstract

The human brain has evolved to have extraordinary capabilities, enabling complex behaviors. The uniqueness of the human brain is increasingly posited to be due in part to the functions of primate-specific, including human-specific, long non-coding RNA (lncRNA) genes, systemically less conserved than protein-coding genes in evolution. Patients who have surgery for drug-resistant epilepsy are subjected to extensive electrical recordings of the brain tissue that is subsequently removed in order to treat their epilepsy. Precise localization of brain tissues with distinct electrical properties offers a rare opportunity to explore the effects of brain activity on gene expression. Here, we identified 231 co-regulated, activity-dependent lncRNAs within the human MAPK signaling cascade. Six lncRNAs, four of which were antisense to known protein-coding genes, were further examined because of their high expression and potential impact on the disease phenotype. Using a model of repeated depolarizations in human neuronal-like cells (Sh-SY5Y), we show that five out of six lncRNAs were electrical activity-dependent, with three of four antisense lncRNAs having reciprocal expression patterns relative to their protein-coding gene partners. Some were directly regulated by MAPK signaling, while others effectively downregulated the expression of the protein-coding genes encoded on the opposite strands of their genomic loci. These lncRNAs, therefore, likely contribute to highly evolved and primate-specific human brain regulatory functions that could be therapeutically modulated to treat epilepsy.

## 1. Introduction

The Human Genome Project revealed that the gene sequences that are translated into proteins comprise only a small percentage (<2%) of the total genomic DNA [1]. Subsequent surveys of the human transcriptome further demonstrated that most RNA molecules in human cells do not code for proteins, and that the majority of these are long non-coding RNA (lncRNA) transcripts. LncRNAs are defined as transcriptional products of non-protein-coding genes with lengths greater than 200 base pairs [2]. This definition also excludes classical smaller RNAs, such as essential ribosomal and spliceosomal components. Other classical computational criteria that define lncRNAs include the absence of an open reading frame greater than 100 amino acids and the lack of any known protein or domain homologies in any shorter reading frame that may occur [3]. Rigorous experimental methods, including the integration of protein mass spectrometry [4] and sequencing of ribosome-bound RNAs with whole-transcriptome data, have documented that most lncRNAs do not bind ribosomes in living cells and that, with rare and highly specific exceptions, lncRNAs never correspond to any detection of predicted translation products [4,5].

Although lncRNAs are found throughout all known species, most lncRNAs are not evolutionarily conserved between closely related lineages. This makes lncRNAs frequently lineage-specific, and sometimes species-specific, which has led to the model, increasingly supported by evidence, that lncRNAs, rather than protein-coding genes, may be responsible for the phenotypic differences between higher vertebrates. This is especially true in organs where interspecies distinctions are particularly pronounced, such as the mammalian brain where lncRNAs are highly expressed [6,7]. In particular, the human and mouse lncRNA catalogs are largely non-overlapping, in part because of lineage-specific gene origination events after the primate-rodent divergence [8]. Therefore, the majority of human lncRNAs are now understood to be primate-specific in their evolution and tissue-specific in their expression [9,10,11]. Taken together, this suggests that lncRNAs could have important roles in human brain activity, disease, and evolution [12].

Recently, we determined that the human brain transcriptome is vastly different in fresh human brain tissues compared to postmortem tissues from both normal brains as well as from brains with many neurological disorders [13]. These differences occur due to rapid loss of neuronal genes expression, despite a rapid increase in glial genes expression that correlate with the growth of both astrocytes and microglia in the human brain after death. LncRNAs and microRNAs were markedly less stable in the postmortem interval than messenger RNAs (mRNAs) of protein-coding genes [14]. These findings highlight the importance of studying lncRNAs in fresh (not postmortem) human brain tissue when possible.

Functionally, lncRNAs are known to regulate gene expression through a variety of mechanisms that include but are not limited to epigenetic modification and post-transcriptional RNA interference, commonly associated with natural antisense transcripts (NATs). Unlike microRNAs, lncRNAs can be both positive and negative regulators of their target mRNA expression or stability, necessitating a case-by-case approach to validating them functionally in cell-culture and animal models. In epilepsy, we have previously shown that differentially expressed antisense lncRNAs can regulate the expression of epileptic signaling genes. One of these lncRNAs, -AS1, is a potent endogenous antisense inhibitor of its overlapping protein-coding gene, BDNF, a key driver and effector of spiking [15]. Summarily, lncRNAs have clear potential to play important roles in the underlying mechanisms of human disease [16,17,18], specifically of human epilepsy, by directly regulating the expression of genes and the activity of pathways associated with epilepsy [12,15,19].

Functional genomic studies on fresh human epileptic spiking versus non-spiking brain regions have revealed several pathways, including the Mitogen Activated Protein Kinase (MAPK) signaling pathway, which have been further implicated mechanistically in epilepsy in human tissues and an animal model [20,21]. In neurons, the MAPK pathway is of critical importance for dendritic spine stabilization and long-term potentiation [22,23]. This suggests that MAPK contributes or reinforces synaptic mechanisms leading to the development and maintenance of epileptic activities and thus may be a therapeutic target for treatment [24,25]. Consistently, inhibitors of MAPK can prevent the development of epileptic spiking in animals [21]. This suggests that methods that inhibit or target the MAPK signaling pathway, including those that involve the manipulation of lncRNAs serving as causal effectors in this pathway, may be developed into novel treatments for epilepsy.

The central goal of the present study was to identify lncRNAs that regulate, or are co-regulated with, protein-coding MAPK signaling genes in fresh human epileptic brain regions. This work is a necessary prerequisite to identify potential endogenous lncRNA-mediated mechanisms of MAPK regulation and new lncRNA targets for therapeutic intervention. Through unbiased clustering of differentially expressed genes from freshly isolated human epileptic neocortical tissue samples, we have identified a group of lncRNAs co-regulated with MAPK genes in the human epileptic neocortex. We studied six of these lncRNAs further, based on their abundance and close correlation with MAPK coding genes in an in vitro model of sustained activity using human neuronal-like Sh-SY5Y cells. Of note, four of the six were antisense to protein-coding genes, and the other two were intergenic (lincRNA; long intergenic non-coding RNA). We confirmed that five out of the six lncRNAs are themselves activity-dependent and that three of the four antisense lncRNAs had reciprocal expression patterns relative to their protein-coding gene partners. Some were directly regulated by MAPK signaling and others downregulated the expression of their antisense coding genes. Taken together, these findings present a newly described and precisely defined subset of the coding/non-coding MAPK interactome in the human epileptic brain that could have important regulatory roles in normal brain function and, furthermore, should be explored as potential therapeutic targets in treating epilepsy.

## 2. Results

### 2.1. LncRNAs Are Co-Expressed within the MAPK Signaling Interactome in Human Epileptic Neocortical Tissue

Differential gene expression was identified through microarray analysis by comparing regions of high epileptic activity to regions of low epileptic activity from neocortical tissue samples of seven patients who underwent surgery for drug-resistant epilepsy (Table 1). Previous studies identified 1288 differentially expressed lncRNAs between high and low epileptic activity regions from each patient [12]. Using a clustering algorithm that identifies groups of genes with similar expression patterns, and then visualizes the results using Cytoscape, we compared the expression patterns of these 1288 lncRNA against known MAPK signaling genes. Of note, 608 probes corresponding to 291 genes associated with the MAPK signaling pathway were included, and 65 of these genes were differentially expressed, including EGR1, BDNF, and ARC [20]. This algorithm allowed us to identify lncRNAs that are co-regulated with MAPK-pathway protein-coding genes, as well as those with potentially important regulatory roles in this important signaling pathway.

Figure 1 shows the composite expression pattern of the lncRNA (green) and co-regulated MAPK signaling protein-coding genes (red) from human epileptic brain tissues. Two hundred thirty-one differentially expressed lncRNAs had similar expression patterns to 42 of the differentially expressed MAPK signaling genes. Interestingly, the majority of the statistically significant differentially expressed lncRNAs were upregulated (Figure 1A) in epileptogenic high-spiking, relative to normal, neocortical tissue samples, with very few downregulated; furthermore, upregulated lncRNA and protein-coding genes were found in the same networks to some extent, while downregulated lncRNAs, with only two exceptions, formed their own co-expression networks where all of the lncRNA nodes were devoid of edges connecting to protein-coding genes (Figure 1B). Of the 231 lncRNAs, we chose six upregulated lncRNAs for further study, based on their annotations, known relationships to protein-coding genes, and relatively high expression levels (Table 1). Four of these lncRNAs are known NATs, and two are lincRNAs.

NATs can upregulate as well as downregulate (in a gene-specific fashion) the expression of their cognate sense protein-coding genes that are expressed from the same genomic loci [26]. This is achieved both epigenetically and post-transcriptionally by a variety of mechanisms [26]. Hence, they have easily defined specific protein-coding targets and are posited to have important MAPK regulatory effects. The four NAT lncRNAs were antisense to the following four protein-coding genes: B-cell translocation gene 3 (BTG3), IQ Motif Containing with AAA Domain 1 (IQCA1), endoplasmic reticulum aminopeptidase 1 (ERAP1), and HECT, C2 and WW Domain Containing E3 Ubiquitin Protein Ligase 2 (HECW2). Interestingly, mutations in and deletions of BTG3 and HECW2 have been associated with cases of human epilepsy and developmental delay [27,28,29]. Of note, BTG3 has also been associated with neurogenesis and HECW2 functions] to stabilize tumor protein p73 (TP73), a member of the family of p53 transcription factors [30].

Taqman qPCR was used to independently validate the microarray-measured differential expression of these lncRNAs. Similar to the microarray results, five of the six differentially expressed lncRNAs had increased expression in areas of high epileptic activity (high spike) compared to areas of low epileptic activity (low spike) (Figure 2). The lncRNA AK023739 had the greatest increase with epileptic activity, demonstrating a seven-fold increase in high spiking regions.

### 2.2. Activity-Dependent MAPK Signaling Increases the Levels of Specific lncRNAs In Vitro

To understand the role of brain activity in high spiking brain regions on lncRNA expression, we repeatedly depolarized the human neuronal-like SY5Y cells with 100 mM KCl (Figure 3) [12]. The four-, eight-, and 24-hour collection points following repeated depolarizations were based on previous studies that demonstrated sustained CREB/MAPK activation at these times [12]. While use of the SY5Y cell line has its limitations, it is important to highlight that using a human cell line is critical for the study of lncRNA, as many lncRNAs are either not conserved in evolution or demonstrate species-specific expression patterns [6]. In this model, EGR1, a downstream MAPK signaling gene whose expression is known to be induced by electrical activity, is dramatically elevated following repeated depolarizations and is used as a positive control for activity-dependent signaling (Figure 3B) [12,25].

Four of the six lncRNAs were significantly increased following repeated depolarizations, but with different kinetics, over 24 hours (Figure 3). The greatest increase was observed with AK023739, which demonstrated a 4.5-fold increase in expression at eight hours following depolarization (Figure 3D). This cell-culture-based finding strongly complements the results from the human epileptic neocortical tissue samples, where AK023739 demonstrated the greatest increase in expression activity (Figure 2). BC028229, CR615000, and BC039550 also demonstrated significant changes in expression following depolarization, increasing three-fold, two-fold, and three-fold, respectively (Figure 3). While BC018494 had a moderate increase in expression at eight hours, AL833303 had no change in expression following depolarization.

We next asked whether MAPK signaling is required for the regulation of these activity-dependent lncRNAs, using the MEK inhibitor PD184352 (10 mM) (Figure 4). In the same depolarization model, we found that MEK inhibition reduces the activity-dependent expression of EGR1, as expected, and AK023739 after four hours. This suggests that the expression of the lncRNA AK023739 is both activity-dependent and requires MAPK signaling to induce its expression. While no statistically significant changes in expression were seen with the other five lncRNAs, AL833303 was downregulated and BC039550 was upregulated with MEK inhibition.

### 2.3. Antisense MAPK lncRNAs Downregulate Their Overlapping Protein-Coding Genes

Antisense lncRNAs are well-known to downregulate the expression of their overlapping protein-coding genes, post-transcriptionally through RNA-RNA interactions, and via several other mechanisms [26]. We therefore compared the expression patterns of the four antisense lncRNAs and their (same-locus-encoded) cognate sense protein-coding genes with repeated depolarizations. Of these, all lncRNAs except CR615000 had a reciprocal pattern of expression with their antisense protein-coding genes (Figure 5), suggesting that for these three lncRNAs the antisense lncRNA may downregulate the expression of their overlapping protein-coding genes.

To confirm the potential regulation of protein-coding gene expression by its antisense lncRNA, we used siRNA to knock down the expression of these four antisense lncRNAs and measured the relative expression of both the lncRNA and its antisense protein-coding gene mRNA, and protein levels for BTG3 and HECW2 (based on antibody availabilities). While siRNA knock-down of BC018494 resulted in the greatest increase in mRNA, both BC028229 and AK023739 knockdown led to increases in the relative expression of BTG3 and HECW2 protein levels, respectively, on Western blots (Figure 6A–D). While further validation studies are required, this suggests that these two lncRNAs may inhibit the expression of their antisense protein-coding genes in neuronal-like cells. This result is consistent with our previous finding that the lncRNA BDNF-AS1 is a negative regulator of the BDNF gene in epilepsy [12]. Unfortunately, antibodies were not available at the time of experimentation for the other two protein-coding genes. However, similar results were observed with siRNA knockdown of BC018494 (Figure 6E), and CR615000 (Figure 6F).

## 3. Discussion

Upregulation of the MAPK signaling pathway is involved in the generation of epileptic brain activity, and inhibition of this pathway can prevent the development of epileptic activity in animals [20,21,24]. This highlights a potential role for MAPK signaling in the treatment of epilepsy. Several lncRNAs are known to be regulated by the MAPK signaling pathway, and in turn, have also been found to regulate MAPK signaling genes [12,31]. This suggests that there is a direct functional relationship between MAPK signaling and lncRNA expression.

Previous studies, outside of the brain, have identified expression networks containing MAPK-pathway signaling genes and correlated lncRNAs [32,33,34]. Given that the expression of most lncRNA is highly tissue-specific, [35] characterizing the activity-driven lncRNA-MAPK interactome in the human brain, as we have done in this study, is a major step in identifying novel regulatory networks involving brain-specific lncRNA. Moreover, as lncRNAs tend to contain more sequences of young evolutionary origin (relative to mRNAs), human lncRNA expression patterns are best studied in humans [6,36]. While most human brain tissues studies use postmortem brain samples, the use of fresh human brain tissues is critical for accurate assessments of human brain transcription, due to the rapid degradation of RNA, which disproportionately affects non-coding RNAs postmortem [13].

Building upon previous findings in human epileptic neocortical tissue, [12] we identified activity-dependent lncRNA genes that co-regulate with MAPK signaling genes in epilepsy using human neocortical brain tissues mapped to precise brain locations from in vivo electrical recordings [24]. To document dynamic changes of these genes, we focused on six lncRNAs, four of which were antisense to protein-coding genes, whose expression was also differential and correlated directly or inversely with their sequence-cognate NAT lncRNAs (Table 1).

MAPK signaling is increased in brain regions with elevated levels of epileptic activity. This increase in MAPK signaling appears to be layer 2/3-specific and is a biomarker for epileptic activity [20,21]. As part of the MAPK-lncRNA interactome, the six identified lncRNAs demonstrated expression patterns similar to previously described MAPK signaling genes in human epilepsy (Figure 1, Table 1). Four of these six lncRNAs were validated in-vitro and showed an increase of expression following an increase in activity dependent signaling (Figure 2 and Figure 3). Moreover, AK023739 appears to be regulated by MAPK-dependent, activity-dependent signaling (Figure 4). These findings suggest that the identified lncRNA, in particular AK023739, demonstrate similar expression patterns to MAPK signaling genes in human neocortical epilepsy and therefore are new lncRNA functional biomarkers for epileptic activity.

LncRNA are increasingly being considered, and starting to be developed, as potential therapeutic targets for neurological diseases as they are highly expressed in the brain and have been shown to contribute to the pathogenesis of Alzheimer’s Disease, Parkinson’s Disease, and epilepsy [7,36,37]. In epilepsy, lncRNAs are differentially expressed and have been demonstrated to regulate gene expression, particularly within the aberrant MAPK signaling pathway [19,38]. In this study, we have newly identified four antisense lncRNAs, three of which demonstrate activity-dependent reciprocal expression patterns with their nearby protein coding genes (Figure 5). These regulatory lncRNAs could be modulated to alter activity-dependent gene expression that may drive the development or maintenance of epileptic circuits and could be considered therapeutic targets.

In accordance with previous studies, here we present evidence to further support the contention that the lncRNA BC028229 regulates BTG3 mRNA and protein expression (Figure 6A,B) [39]. Moreover, we have also demonstrated that the lncRNA AK023739 may have a regulatory effect similar to the nearby protein coding gene HECW2 (Figure 6C,D). Consistently, deletions and mutations in BTG3 and HECW2 have been associated with developmental delay and epilepsy [28,29]. This suggests that the lncRNA, by regulating expression of overlapping protein-coding genes, may have a role in epileptic activity.

One way to target lncRNAs is through RNA interference with small interfering RNA (siRNA). There are currently four FDA-approved medications, including Inclisiran (a siRNA that treats hypercholesterimia by suppressing the PCSK9 gene), that harness siRNA technology, with at least six in Phase 3 Clinical Trials [40,41]. Additional RNAi-based therapeutics have been approved by the EMA, and dozens of siRNA-driven drug candidates are in clinical trials in advanced non-US jurisdictions. Targets of these siRNA therapeutics are diverse and span clinical themes ranging from lowering cholesterol to treating ischemic optic neuropathy [40]. This increasing reliance on RNAi-based therapeutics in our post-genomic era highlights the safety and efficacy of RNA interference technologies, and of the sequence-based drugs that utilize them in the treatment of human disease.

Remarkably, recent studies have demonstrated success in delivering siRNA across the blood-brain barrier, highlighting the potential to treat neurological diseases such as epilepsy with sequence-based, RNAi-driven drugs that suppress the molecular causes of epileptogenic spiking [42]. While the majority of these trials focus on targeting protein-coding genes, lncRNAs are considered viable drug targets, and studies where lncRNAs are being targeted with sequence-based therapies are ongoing [43]. Since lncRNA genes typically have a limited number of network interactions as nodes (in contrast to protein-coding genes), are connected only by limited and sparse edges to the rest of their regulatory networks (due to their young evolutionary origins resulting in insufficient time for embedding deeper into these networks, as well as to the sequence-specificity of their RNA-RNA interactions), and are highly tissue-specific, treatments that target lncRNAs are posited to exhibit more specificity with fewer off-target effects, and therefore represent an exciting class of future therapeutics.

## 4. Materials and Methods

### 4.1. Electrically Mapped Human Epileptic Neocortical Tissues

Human epileptic neocortical tissue samples were obtained from 7 patients, ranging in age from 3 years to 33 years, who underwent a two-staged surgery for drug resistant epilepsy. Samples were received following informed consent as part of a research protocol that was approved by an Institutional Review Board (IRB) at Wayne State University and at the University of Illinois–Chicago. Patients first underwent invasive EEG recording with electrodes placed directly on the neocortical surface to obtain the most precise electrical profile (surface electrical activity map) of the epileptic brain. Using the EEG grid recording and cortical mapping, the epileptic focus was then identified and removed. No additional tissue was removed for these studies; however, the *en bloc* resections yielded samples ranging from high to no epileptic activity. Once extracted, each tissue sample was divided so that half was immediately frozen on dry ice and stored at −80 °C for molecular analysis while the other half was fixed in 4% paraformaldehyde (PFA) and later embedded in an optimal cutting temperature compound for histological analysis. Each tissue sample was identified by the intracranial grid EEG electrode number corresponding to specific electrical features and location on the brain. Spike frequency was defined as the number of epileptic spikes in 10 minutes, averaged together from three independent ten-minute time periods using an automated algorithm as described [20]. All data and tissue are stored with de-identified information in the University of Illinois NeuroRepository.

### 4.2. Microarray and Gene Clustering

To ensure that only gray matter was included, RNA was extracted from even amounts of neocortical layers I–VI of the brain regions underlying each electrode using the RNeasy Qiagen Lipid Kit (Qiagen, MD, USA) according to manufacturer’s instructions. A dye-flip quadruplicate, two-color microarray analysis was then performed to detect unique genes in areas of the resected epileptic neocortex defined as having high epileptic activity based on EEG recordings and compared with low or no activity brain regions within each patient. Custom-designed lncRNA (based on manually curated data) and protein coding microarrays were performed as previously described using the Agilent Technologies platform [12].

For this study, we focused on differentially expressed lncRNAs and protein-coding genes that were previously known, from gene-ontology analyses, to be MAPK-pathway signaling genes. From 7 patients comparing high and low spiking brain regions, the total number of differentially expressed lncRNAs was 1288 genes and the total number of differentially expressed MAPK-pathway protein-coding genes was 65. The probe expression profiles across the 16 samples were clustered in ´R´ using the Pearson correlation (*p*-value, *p* < 1 × 10^−5^) and visualized with Cytoscape. The plugin AllegroMcode was used with default parameters to characterize clusters. This study focused on a cluster that contained a high proportion of both lncRNAs (231) and MAPK signaling genes (42). We used an arbitrary cut-off for genes of interest of absolute fold change >1.3 and a false detection rate <5%. We also focused on genes with stronger fluorescent signals on the microarray, suggesting a high expression level of these genes in the brain.

### 4.3. Cell Culture, Reagents, Depolarizations, and Electroporation

All in vitro experiments were performed using the Sh-SY5Y (SY5Y) cells between 4–16 passages from the ATCC. Cells were grown in high glucose Dulbecco´s Modified Eagle Medium (DMEM) (Gibco, Grand Island, NY, USA) supplemented with 10% Fetal Bovine Serum (FBS) (Gibco) and 1% Penicillin-Streptomycin (PS) (Sigma, St. Louis, MO, USA). Cell lines were maintained at 37 °C with 5% CO_2_. Repeated depolarizations were performed using 100mM potassium chloride (KCl) in DMEM added to the SY5Y cells for 5 min and then replaced with DMEM every two hours to create a sustained, activated MAPK signaling profile, as previously described [12]. MAPK inhibition was achieved using the MEK inhibitor PD184352 (PD18) at a final concentration of 10mM in DMSO.

For transfections, cells were electroporated using the Neon electroporation apparatus using the following parameters: 1200 V, pulse number 2, pulse width 20. Said parameters were based on previously described settings in the SY5Y cell line [12]. Silencer® select small interfering RNA (siRNA) targeting the lncRNAs BC028229, AK023739, CR615000, BC018494, and control siRNA were purchased from ThermoFisher and used at a final concentration of 100nM.

### 4.4. RNA Extraction, Reverse Transcription, and qPCR

RNA was extracted from the SY5Y cells using the RNeasy Qiagen MiniKit (Qiagen, Germantown, ML, USA) according to the manufacturer’s instructions. All reverse transcription reagents were purchased from Invitrogen (Thermo Fisher Scientific, Waltham, MA, USA). For quantitative real-time reverse transcription PCR (qRTPCR; qPCR) reactions, Taqman Gene Expression Master Mix (Thermo Fisher Scientific, USA) was used in combination with Taqman primer probes (Thermo Fisher Scientific, USA) for glyceraldehyde 3-phosphate dehydrogenase (GAPDH) (Hs02786624_g1), EGR1 (Hs0152928_m1), BTG3 (Hs00199064_m1), HECW2 (Hs01372347_m1), IQCA1 (Hs00226839_m1), BC018494 (Hs03657050_s1), AK023739 (Hs04404078_m1), CR615000 (Hs03861783_g1), AL833303 (Hs03842996_s1), and BC039550 (Hs00295184_m1). Custom Taqman primer probes were made against BC028229 (Thermo Fisher Scientific, USA).

### 4.5. Western Blots

Whole-cell extracts were lysed in RIPA buffer (SigmaAldrich, Waltham, MA, USA) supplemented with 1% Halt Protease Inhibitor. Protein was quantified using the Bradford reaction. 10μg of each sample was loaded into individual lanes and subjected to sodium dodecyl-sulfate polyacrylamide gel electrophoresis (SDS-PAGE), followed by transfer onto a polyvinylidene difluoride (PVDF) membrane (Millipore, Billerica, MA, USA). Membranes were incubated overnight in primary antibodies against BTG-3 (1:1000) (Abcam, UK), HECW2 (1:1000) (SigmaAldrich, USA), and actin (1:3000) (Cell Signaling, USA) at 4 °C in 5% milk in 1% tween in tris buffered saline. Horseradish peroxidase (HRP) conjugated goat anti-mouse IgG or anti-rabbit IgG secondary antibodies (BioRad) were used, and the signal was detected using Thermo Scientific SuperSignal West Pico Chemiluminescent Substrate (Thermo Fisher Scientific, USA).

### 4.6. Statistical Analysis and Experimental Design

All statistical analyses were performed using Graph Pad Prism 9.0. When comparing two groups, *t*-tests were performed. When comparing three groups, multiple time-points, or treatments, one-way analysis of variance (ANOVA) or two-way ANOVA with repeated measures (RM) was performed. To account for multiple comparisons, the Sidak test for multiple comparisons was performed. For test specifics, please see the results section and figure legends. The *n* stands for the sample size and each *n* is specified in the results section. All error bars are displayed as standard error of the mean (SEM). Significance was designated as *p* < 0.05. For experiments that involved the application of compounds, vehicle controls were always included.

## Figures and Tables

**Figure 1 ncrna-09-00003-f001:**
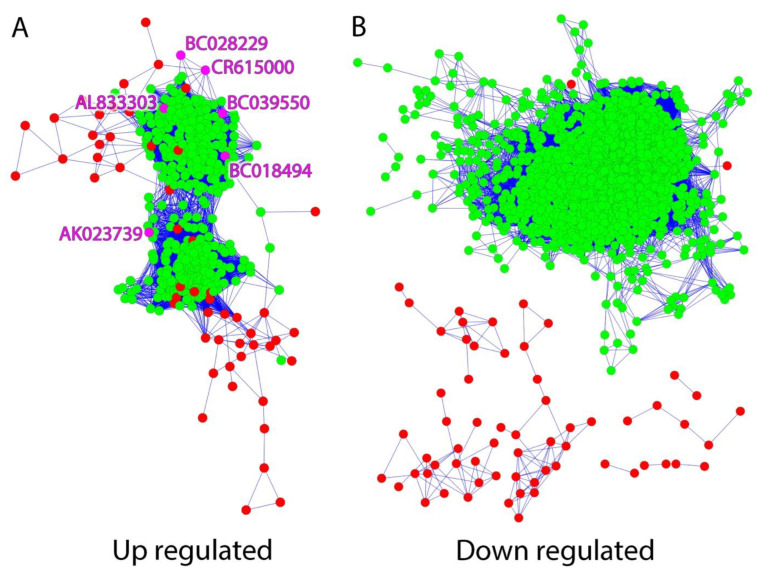
MAPK-pathway signaling genes and specific lncRNAs are co-differentially expressed in an activity-dependent fashion in human epileptic brain tissues. Gene clustering of microarray results from seven patients with neocortical epilepsy demonstrates that certain lncRNAs (green) have significantly similar expression patterns to the MAPK-pathway signaling genes (red). Each node (green, red or magenta circles) corresponds to a gene and each link between nodes correspond to a Pearson correlation p-value *p* < 0.00001 (R > 0.9, 14 samples), indicating that the closer two linked nodes are to each other, the more closely their expression patterns resemble each other. LncRNAs of interest (magenta) were identified and subsequently labeled. (**A**) Clustering of MAPK-pathway signaling genes with upregulated lncRNAs shows 231 significantly upregulated lncRNA and 42 MAPK signaling genes with a similar expression pattern (fold change > 1.3, false discovery rate < 5%). (**B**) Clustering of MAPK-pathway signaling genes with downregulated lncRNAs shows that the MAPK-pathway signaling genes does not cluster with the down regulated lncRNAs..

**Figure 2 ncrna-09-00003-f002:**
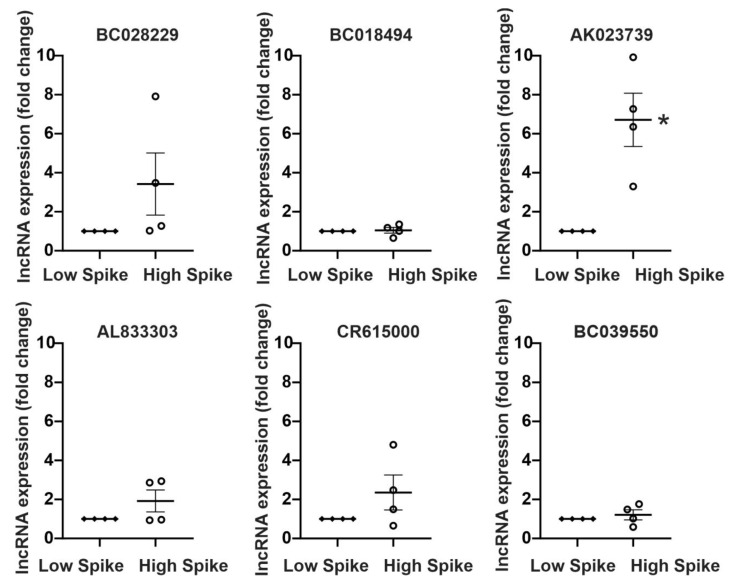
qPCR of human epileptic neocortical tissue samples confirms the differential expression of lncRNAs. Within each patient, brain regions that were previously identified as having high epileptic signaling (High Spike) or low epileptic signaling (Low Spike), were compared. Five of the six lncRNAs demonstrated increased expression in areas of high epileptic spiking activity, including a 3.7-fold increase in BC028229, a seven-fold increase in AK023739 (* *p* < 0.05, one-sample *t*-test, *n* = 4), a two-fold increase in AL83303, a three-fold increase in CR615000, and a 1.2-fold increase in BC039550. No change in expression was observed for the lncRNA BC018494.

**Figure 3 ncrna-09-00003-f003:**
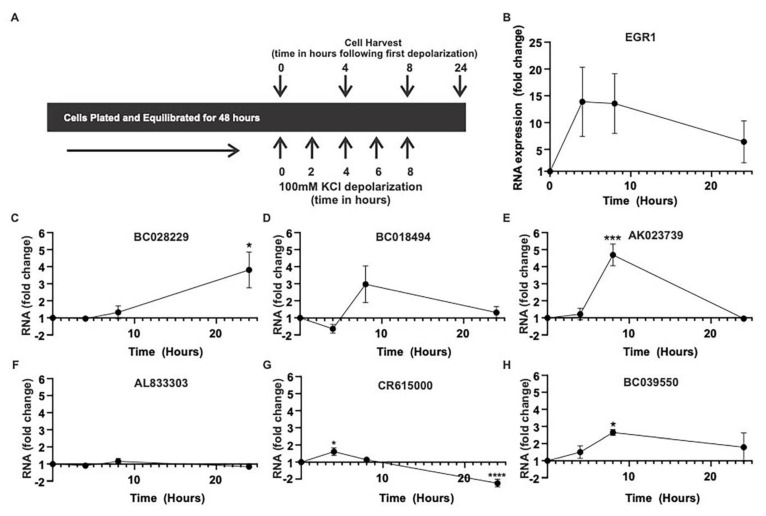
Variability in activity-dependent lncRNA induction. (**A**) Repeated KCl-induced depolarizations were used to model activity-dependent signaling in vitro. qPCR was performed to measure RNA expression at the specified time points. All results are displayed as fold change in comparison to time-matched controls. (**B**) EGR1 increases with activity at four- and eight-hours following depolarization. (**C**) BC028229 shows a steady increase in activity-dependent expression reaching three-fold by 24 hours (* *p* < 0.05, two-way ANOVA, *n* = 3, DF = 6), (**D**) BC018494 increases expression by three-fold at eight hours (*p* = 0.12, two-way ANOVA, *n* = 3, DF = 6), (**E**) AK023739 increases 4.5-fold at eight hours (*** *p* = <0.001, two-way ANOVA, *n* = 3, DF = 6), (**F**) AL83303 does not change, (**G**) CR61500 has a two-fold increase at four hours (* *p* < 0.05, **** *p* < 0.0001, two-way ANOVA, *n* = 3, DF = 6), and (**H**) BC039550 has a three-fold increase at eight hours (* *p* < 0.05, two-way ANOVA, *n* = 3, DF = 6).

**Figure 4 ncrna-09-00003-f004:**
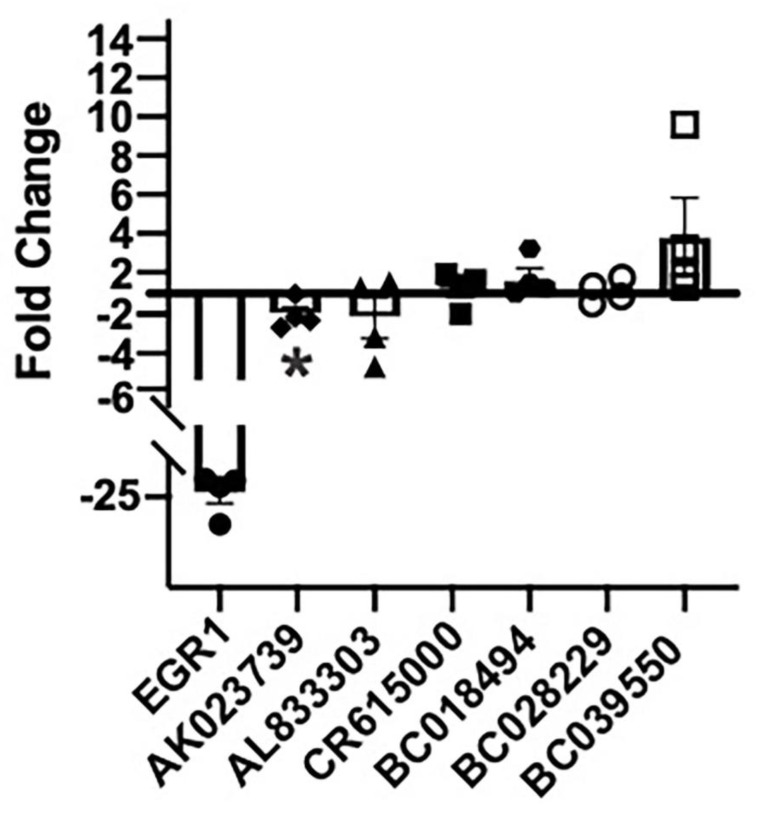
MAPK signaling modulates AK023739 expression following cell depolarization. 10 M PD18 significantly reduced the activity-dependent expression of AK023730 following 100mM KCl depolarization at four hours via qPCR (* *p* < 0.05, one sample *t*-test, *n* = 4) Results are displayed as fold change, comparing depolarization with MEK inhibition to depolarization with vehicle control, but had variable effects on the other lncRNAs. The decrease in EGR1 expression was included as an internal control.

**Figure 5 ncrna-09-00003-f005:**
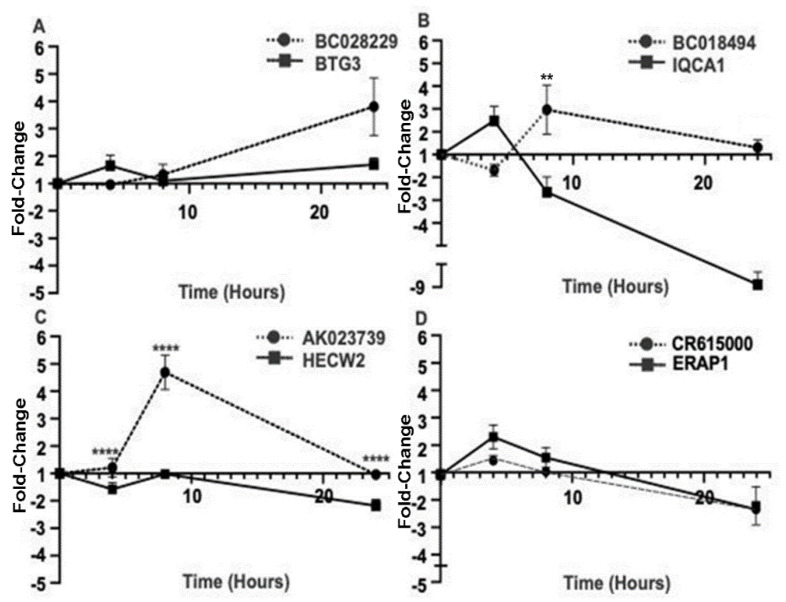
Sense-antisense coding/non-coding gene pairs demonstrate reciprocal expression patterns after repeated depolarizations. (**A**) Repeated KCl depolarizations in the Sh-SY5Y cells resulted in significantly different expressions of BC028229 and BTG3 at 24 hours (*p* = 0.06, two-way ANOVA, *n* = 3, DF = 14), (**B**) BC018494 and IQCA1 at 24 h (** *p* < 0.01, two-way ANOVA, *n* = 3, DF = 14), and (**C**) AK023739 and HECW2 at four, eight, and 24 h (**** *p* < 0.0001, two-way ANOVA, *n* = 3, DF = 14). All results are displayed as fold changes in comparison to time matched control. (**D**) CR615000 and ERAP1 demonstrate similar patterns of expression with no significant changes between the two expression patterns.

**Figure 6 ncrna-09-00003-f006:**
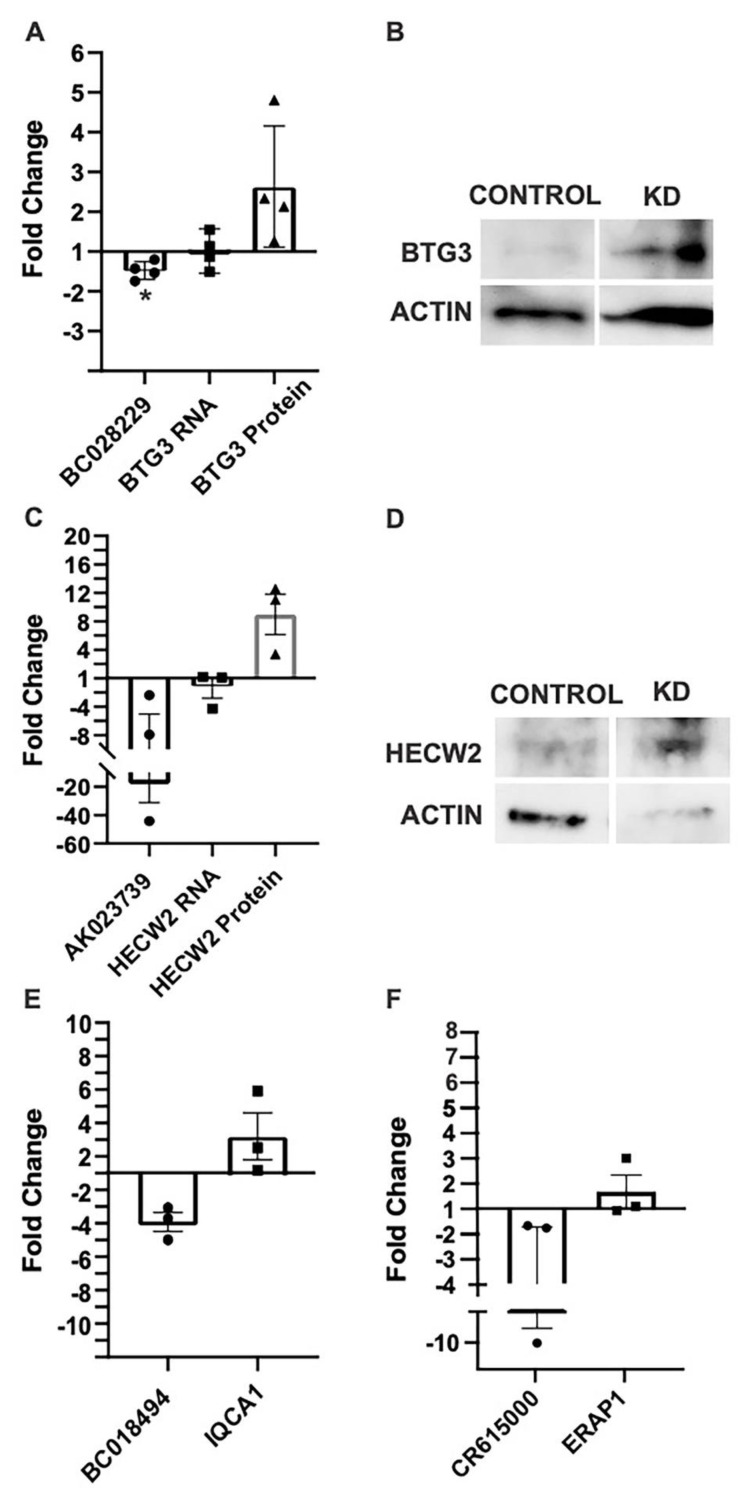
Antisense lncRNAs negatively modulate the expression of their overlapping protein-coding genes in vitro. (**A**) siRNA knockdown of BC028229 in Sh-SY5Y cells results in a significant decrease in BC028229 (* *p* < 0.05 one sample *t*-test, *n* = 4) and an increase in the expression of BTG3 mRNA via qPCR and (**B**) an increase in the expression of BTG3 protein via Western blot (*p* = 0.12, one sample *t*-test, *n* = 4). (**C**) siRNA knockdown of AK023739 results in a decrease in AK023739 and (**D**) an increase in the expression of HECW2 protein, as shown by Western blot (*p* = 0.12, one sample *t*-test, *n* = 4). (**E**) Knockdown of BC018494 results in a significant decrease in BC018494 (*p* < 0.05, one sample *t*-test, *n* = 3) and an increase in the expression of IQCA1 mRNA, as shown byqPCR (mean = 3.27, *p* = 0.25, one sample *t*-test, *n* = 3). (**F**) CR61500 knockdown results in no significant changes in ERAP1 expression. All results are a fold change in comparison to mock transfection control.

**Table 1 ncrna-09-00003-t001:** Upregulated lncRNA genes identified as co-expressed with MAPK signaling genes. Following detailed annotation, six upregulated lncRNA were chosen as candidates for further analysis.

lncRNA	Fold Change (FC)	False Discovery Rate (FDR)	lncRNA Classification	Overlapping Coding Gene	Mutations Associated with Epilepsy
BC028229	2.1	0.00052	antisense	BTG3	Yes
BC018494	1.7	1.8	antisense	IQCA1	No
AK023739	1.7	3.2	antisense	HECW2	Yes
AL833303	1.5	0.00025	intergenic	N/A	No
CR615000	1.5	3.9	antisense	ERAP1	No
BC039550	1.3	0.03	intergenic	N/A	No

## Data Availability

The data presented in this study are available on request from the corresponding author.

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
