# Peer review of "Activity-Dependent Non-Coding RNA MAPK Interactome of the Human Epileptic Brain"

_ncrna, 2023, doi:10.3390/ncrna9010003_

Round 1

Reviewer 1 Report

In this manuscript the authors have analyzed six long non-coding RNAs (lncRNAs) upregulated in brain tissue of high activity in epilepsy patients based on micro-array data. Upregulation was validated by qPCR for 5/6 lncRNAs and 3/6 were also found significantly increased at different time-points following repeated KCl depolarization in SH-SY5Y cells. In this model the authors also show MEK inhibition reduced expression of lncRNA AK023739 and some reciprocal expression was observed between 4 antisense lncRNAs and their corresponding protein-coding genes. Finally, siRNA KD of the 4 antisense lncRNAs lead to upregulation of corresponding genes either at the protein level or at the mRNA level. 

            While activity-dependent modulation of lncRNAs is of interest and convincing evidence of a link between lncRNA AK023739 and MAPK signaling is presented the novel findings from this study are limited. The paper could benefit from additional functional analyses and improved presentation of the current results. 

Main points:

1.    The writing/presentation of the first results section is somewhat misleading. While the previous study where this data was first described is referenced (Lipovich et al), the inclusion of the tissue collection and microarray analysis in the methods and the description of these in the results implies this is newly generated data. What has been done previously should be clearly stated and only briefly summarized. Methods describing how previously published data was generated and analyzed should be cut. After referencing/summarizing the data being employed here the authors should clarify exactly what new analysis has been done here and why. A clustering analysis of this data was also previously presented (Lipovich et al) and its not obvious if this new clustering analysis is different and if so how and why this new clustering was performed.

2. Statistical results are missing from figure 2. Have statistical tests been done? Did none reach significance for all results presented in these figures? It appears that qPCR showing upregulation of at least AK023739 in high spike samples might be significant. If no results are significant this should be stated and/or increases of expression referred to as only as trends detected. This would also suggest that these experiments were underpowered. While patient tissue samples might be understandably limited only 4/8 (presumably available) were utilized, why is this?

 3. Western blot/qPCR experiments presented in figure 6 seem to be underpowered. More replicates should be added, otherwise, the current conclusions that KD of lncRNAs increase protein-coding gene levels are overstated and it should be clear that only trends were detected. Also the authors seem to suggest that KD of both BC018494 and CR615000 increase mRNA levels of IQCA1 and ERAP1, respectively (lines 334/335). This is problematic given (a) that the authors report no-significant increase of ERAP1 in the figure legend and (b) increase of IQCA1 was also not significant (p=0.25).

4. Given the above I would recommend that the authors perform further validation experiments to confirm whether lncRNAs regulate their associated protein-coding genes. For example, detection of protein-coding gene expression following ectopic expression or over-expression of lncRNAs in the model system.

5. Overall, the study provides limited novel findings and could be greatly improved by additional functional validations e.g. functional validation of the effects of KD/overexpression of AK023739 which appears to be the most promising candidate.

Minor points

1.    Limitations of the SH-SY5Y model should be discussed.

2. Literature review should be more thorough, some highly relevant citations appear to be missing e.g. (Rahmani, Z, 2006, J Cell Sci).

3. Both the Introduction and Discussion are overly long and detailed. Should be more succinct/focused on the most relevant information.

4. The representation of replicates in figures with different symbols for different groups adds unnecessary complexity since data is already plotted separately by group.

Reviewer 2 Report

In this study, Kirchner et al have reported a newly described coding/non-coding MAPK interactome in the human epileptic brain that could have important regulatory roles in normal brain function. The main aim of the study is interesting, the experimental design is accurate, and the results and discussion sound well.

There are some concerns to be replied for improving the final version of the manuscript.

In the results section, it is necessary to present the outputs of statistical tests; for example, F values and degrees of freedom of ANOVAs.

Can the authors explain why they have examined activity-dependent lncRNAs expression in the Sh-SY5Y cells at 4, 8, and 24 hours after KCL depolarization?

The y axes of Figure 5 are lacking a title and/or unit.

In figure 6 D, I think that the increased level of HECW2 protein is due to a decrease in actin protein. Please explain this issue. Also, it would be better to show expression of lncRNAs and proteins of the control group in each graph with an independent bar.
